# A Sparse Recovery Algorithm Based on Arithmetic Optimization

**Qingfeng Zhang [1], Dong Hu [1,2], Chao Tang [1,2] and Jufang Xie [1,2,*]**

1   College of Engineering and Technology, Southwest University, Chongqing 400715, China
2   International Research and Development Center of Smart Power Grid and Equipment New Technology,
    Southwest University, Chongqing 400715, China
*   Correspondence: xiejufangswu@163.com

**Abstract:** At present, the sparse recovery problem is mainly solved by convx optimization algorithm and greedy tracking method. However, the former has defects in recovery efficiency and the latter in recovery ability, and neither of them can obtain effective recovery under large sparsity or small observation degree. In this paper, we propose a new sparse recovery algorithm based on arithmetic optimization algorithm and combine the ideas of greedy tracking method. The proposed algorithm uses arithmetic optimization algorithm to solve the sparse coefficient of the signal in the transform domain, so as to reconstruct the original signal. At the same time, the greedy tracking technique is combined to design the initial position of the operator before solving, so that it can be searched better. Experiments show that compared with other methods, the proposed algorithm can not only obtain more effective recovery, but also run faster under general conditions of observation number. At the same time, It can also recover the signal better in the presence of noise.

**Keywords:** signal processing; compressed sensing; sparse recovery algorithm; arithmetic optimization algorithm; greedy tracking method





## 1. Introduction

Mechanical vibration is a common phenomenon in machinery and equipment. Since it contains rich information about the operation of the equipment, monitoring and extracting useful information from the vibration process can help people to better diagnose and monitor the condition of the equipment [1]. However, the mechanical vibration signal is a dynamic complex non-stationary signal with high frequency [2]. To achieve undistorted restoration, the sampling technique based on Nyquist sampling theory is often used, which requires that the sampling frequency must be higher than twice the highest frequency of the signal, otherwise the original signal will not be accurately reconstructed. In recent years, the use of large equipment is increasingly diverse, and its operation process also produces more complex changes, such as equipment clearance, vibration conditions, friction, collision, randomness of frequency, etc. The data generated by mechanical equipment is also developing towards the direction of "big data" [3]. If we still use traditional Nyquist sampling law for sampling at this time, it will inevitably require higher sampling frequency and produce a huge amount of monitoring data, and the transmission and storage of these data has become the bottleneck problem to be solved urgently.

The emergence of compressed sensing theory [4] (CS) better solves the above problems, which can sample the signal at a frequency far less than Nyquist sampling and then reconstruct it accurately. Since the sampling method can sample at much lower than the Nyquist sampling rate, it greatly reduces the sampling speed of the device and avoids collecting a large amount of useless data, saving data storage space and reducing signal processing time. Therefore, compressed sensing is widely used in image compression [5,6], medical imaging [7,8], communication system [9] and many other fields [10,11]. The recovery algorithms of compressed sensing model mainly include greedy algorithm and convex optimization algorithm [12]. When no noise exists, the greedy algorithm and its optimization algorithm

need too many measurements and have low recovery accuracy, which cannot guarantee the global optimal solution. The convex relaxation algorithm has high precision and requires less observation times, but it has high computational complexity and is easy to produce artificial effects. Computational intelligence method is an effective modern intelligence method to solve combinatorial optimization problems. Literature [13] applies the hybrid simulated annealing algorithm to the solution of compressed sensing, which improves the reconstruction accuracy of images. Literature [14] applied it to SAR high-resolution range image reconstruction based on genetic algorithm combined with compressed sensing, and this method can reconstruct SAR scene targets with fewer measurements. Literature [15] applied the forbidden search algorithm to DOA estimation, and the method was able to obtain the global optimal solution and reduce the computational effort. Literature [16] combines compressed sensing and convolutional neural network to propose an open-circuit fault diagnosis method for photovoltaic inverters. Literature [17] propose a method of optical fiber sensing signal processing based on Compressed Sensing (CS) to improve the accuracy of vibration location information of the **Φ**-OTDR system. Although there are many research algorithms for sparse recovery of compressed sensing models, few researches are focused on mechanical vibration signals [18,19].

Aiming at these problems, for accurate recovery machinery vibration signal, this paper propose a reconstruction method based on arithmetic optimization algorithm and combined with greedy algorithm pruning technique. The proposed algorithm uses arithmetic optimization algorithm to solve the sparse coefficient of the signal in the transform domain, so as to reconstruct the original signal. At the same time, the greedy tracking technique is combined to design the initial position of the operator before solving, so that it can be searched better.

The remainder of this paper is organized as follows. Section 2 briefly introduces the Arithmetic Optimization Algorithm. Section 3 describes Compressed perception theory model and the proposed sparse recovery algorithm based on arithmetic optimization algorithm. Section 4 presents the experimental results conducted using the proposed method. In addition, it compares the proposed method with traditional methods and analyzes the effects of important factors. Finally, Section 5 states the conclusion.

## 2. Arithmetic Optimization Algorithm

The Arithmetic Optimization Algorithm (AOA) was proposed by Laith Abualigah et al. in 2021, which exploits the distributional behavior of the main arithmetic operators in mathematics, including (Multiplication (*M*), Division (*D*), Subtraction (*S*), and Addition (*A*)). The mathematical modeling of AOA is also performed to evaluate the performance, convergence behavior and computational complexity of the proposed object-oriented method under different scenarios. Experimental results show that compared with other 11 common optimization algorithms, this algorithm has better effect and convergence behaviors in solving optimization problems [20].

Consider the minimization problem min $f(x)$, $x \in \Omega$, where $f$ is the fitness function and the set $\Omega$ is the solution space. In the arithmetic optimization algorithm, the candidate solutions cooperate and coexist with each other. Each solution is called an "operator", looking for the best position in the $\Omega$. An operator updates its position through its own "experience" and the "experience" of the surrounding operators in the process of searching. Here, "experience" means memorizing and tracking the best position encountered. The operator position is updated by the following equation:

$$x_{i,j}(t+1) = \begin{cases} best(x_j) \div (MOP + \tau) \times ((UB_j - LB_j) \times \mu + LB_j)), & r2 < 0.5 \\ best(x_j) \times MOP \times ((UB_j - LB_j) \times \mu + LB_j), & otherwise \end{cases} \quad (1)$$

$$x_{i,j}(t+1) = \begin{cases} best(x_j) - MOP \times ((UB_j - LB_j) \times \mu + LB_j), & r3 < 0.5 \\ best(x_j) + MOP \times ((UB_j - LB_j) \times \mu + LB_j), & otherwise \end{cases} \quad (2)$$

Equation (1) is the operator updating strategy adopted in global search, and Equation (2) is the operator updating strategy adopted in local search. When $r1 > MOA$, the operator updates position by Equation (1), that is Division ($D$) and Multiplication ($M$) are used for global search. When $r1 < MOA$, operator updates position by Equation (2), that is Subtraction ($S$) and Addition ($A$) are used for local search. $MOA$ is given by the following equation.

$$MOA(t) = Min + t \times \left( \frac{Max - Min}{M\_Iter} \right) \tag{3}$$

where $MOA(t)$ denotes the function value at the $t$th iteration. $t$ denotes the current iteration, which is between 1 and the maximum number of iterations ($M\_Iter$), where $x_{i,j}(t + 1)$ denotes the $i$th solution in the next iteration, $x_{i,j}(t)$ denotes the $j$th position of the $i$th solution at the current iteration, and $best(x_j)$ is the $j$th position in the best-obtained solution so far. $\tau$ is a small integer number, which is determined in the AOA algorithm. $UB_j$ and $LB_j$ denote to the upper bound value and lower bound value of the $j$th position, respectively. $\mu$ is a control parameter to adjust the search process, which is fixed equal to 0.5. *Min* and *Max* denote the minimum and maximum values of the accelerated function. $r1,r2,r3$ are random numbers. $MOP$ is given by the following equation:

$$MOP(t) = 1 - \frac{t^{1/\omega}}{M\_Iter^{1/\omega}} \tag{4}$$

where $MOP(t)$ denotes the function value at the $t$th iteration, and $t$ denotes the current iteration. $\omega$ is a sensitive parameter and defines the exploitation accuracy over the iterations, which is fixed equal to 5.

## 3. Sparse Recovery Algorithm Based on Arithmetic Optimization Algorithm

Compression perception theory proposes that if a signal itself is sparse, or can be represented as sparse by some transformation basis, it can be observed by a measurement matrix uncorrelated with the transformation basis with a certain number of observations to obtain a set of observations much smaller than the length of the original signal, and then use the correlation recovery algorithm to recover the original signal from the lesser number of observations.

In this section, a sparse recovery algorithm is proposed based on arithmetic optimization and greedy tracking. The following three aspects are carried out from operator position and fitness, initialization and operator position update mechanism. It does not need to adjust many parameters except the population size and stopping criterion, which are standard parameters in all optimization algorithms.

### 3.1. Operator Position and Fitness

The compressed sensing recovery model is the $l_0$ norm of the following equation:

$$\min\|\theta\|_0 \text{ s.t.} y = \Phi x = \Phi \Psi \theta = A\theta \tag{5}$$

where: $x \in R^N$ is the original signal, $\Phi \in R^{M \times N}$ is the measurement matrix, $y$ is the $M$-dimensional observation, $\theta$ is the sparse coefficient expressed by the sparse transform base $\Psi$, $A \in R^{M \times N}$ ($A = \Phi\Psi$) is the perceptual matrix. $\|\bullet\|$ represents the 0 norm, i.e., the number of non-zero elements in the vector. If $\|\theta\| \leq K \leq N$, say $x$ is $K$-sparse on $\Psi$, And say $K$ is the sparsity of $x$. Figure 1 shows the vibration signal of length 512 and its sparse transformation under the DCT orthogonal basis.

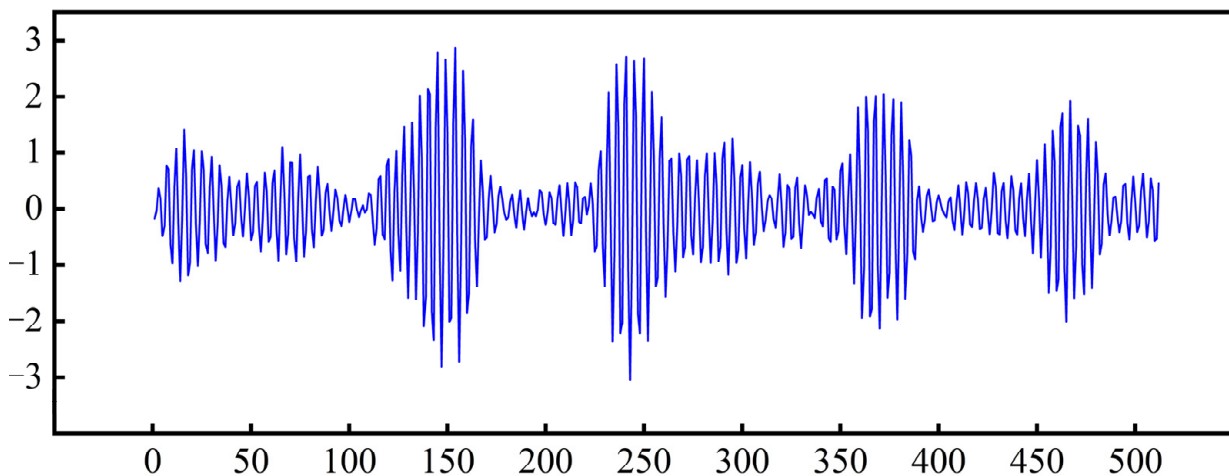

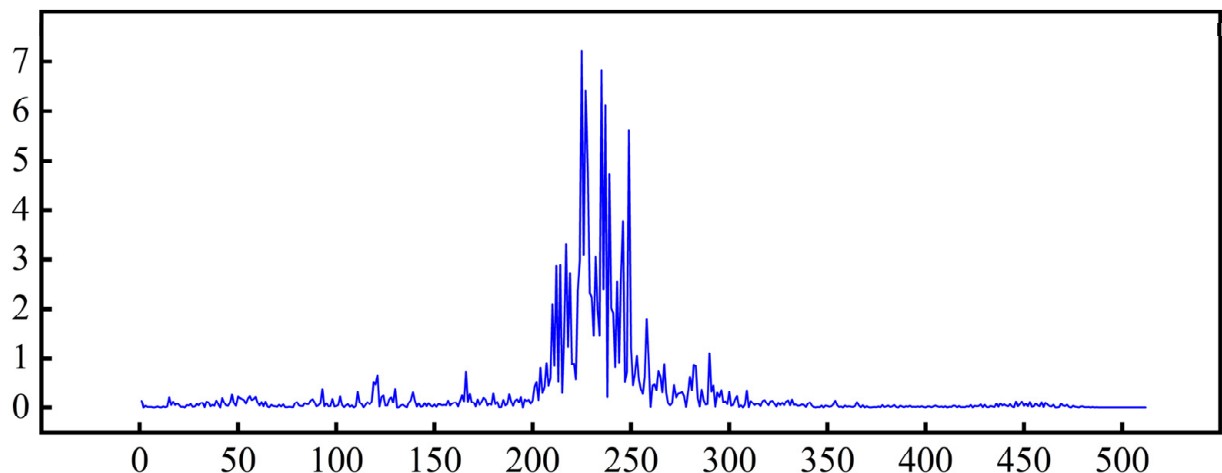

**Figure 1.** Vibration signal and its sparse coefficient.

The set of indicators corresponding to the non-zero elements of the vector $\boldsymbol{\theta}$ is called the support of $\boldsymbol{\theta}$. If the projection $\boldsymbol{\theta}$ of $x$ on $\boldsymbol{\Psi}$ is known to be $K$ sparse, problem (5) can be transformed into its equivalent form [21]:

$$\min_{\boldsymbol{\theta} \in R^N} \|A\boldsymbol{\theta} - \boldsymbol{y}\|_2, \text{s.t. } \|\boldsymbol{\theta}\|_0 \leq K \tag{6}$$

where $\|\bullet\|_2$ stands for the 2-norm. To solve Equation (6), the strategies adopted by many methods can be divided into two steps: the first step is to obtain the position set $\beta$ of the reconstructed signal solution, In the second step, the least square method is used to obtain the original signal estimation solution.

$$\boldsymbol{\theta}_\beta = A_\beta^+ \boldsymbol{y}, \ \boldsymbol{\theta}_{L-\beta} = 0 \tag{7}$$

where, $|\bullet|$ represents the number of elements in the set, and $L = \{1,2,3,\ldots,N\}$ represents the complete set. $\boldsymbol{\theta}_\beta$ and $\boldsymbol{\theta}_{L-\beta}$ respectively represent the components of $\boldsymbol{\theta}$ with $\beta$ or its complement as indexes. $A_\beta$ is the submatrix of $A$ indexed by the elements in $\beta$, and $A_\beta^+ = (A_\beta^\mathrm{T} A_\beta)^{-1} A_\beta^\mathrm{T}$ is its Moore-Penrose generalized inverse. If $\boldsymbol{\theta}$ is the solution to problem 5, then

$$\min_{\|\boldsymbol{\theta}\|_0 \leq K} \|A\boldsymbol{\theta} - \boldsymbol{y}\|_2 = \left\|A_\beta A_\beta^+ \boldsymbol{y} - \boldsymbol{y}\right\|_2 \tag{8}$$

Namely:

$$\beta = \underset{|\beta|=K}{\mathrm{argmin}}\left\|A_\beta A_\beta^+ y - y\right\|_2 \tag{9}$$

This transforms problem (8) into solving the combinatorial optimization problem (9), which is essentially solving the support estimate $|\beta| = K$ for $\beta$.

So in the algorithm proposed in this paper, the operator position is defined as the support estimate $\beta$ of $\theta$. The fitness function of the operator is defined as

$$f(\beta) = \left\|A_\beta A_\beta^+ y - y\right\|_2 \tag{10}$$

Ultimately, problem (6) is transformed into the minimization problem to be addressed:

$$\underset{|\beta|=K}{\min} f(\beta) \tag{11}$$

The algorithm proposed in this paper obtains the position of the operator by solving Equation (11), namely, the support estimation of the signal to be recovered in the sparse domain. Since the signal is sparsely transformed, the estimated solution obtained by Equation (7) is the sparse coefficient $\theta$ of the signal in the transform domain. After obtaining the sparse coefficients by Equation (7), the signal is reconstructed by $x = \Psi\theta$. So far the purpose of solving the sparse recovery problem is reached.

### 3.2. Initialization

Assume the operator size is $Q$ and the initial operator population is denoted as $\{\alpha_i(0) \mid i = 1, 2, \ldots, Q\}$. The following is the initialization of the operator.

The first operator: the operator position $\alpha_1(0)$ is set to the set of indicators corresponding to the largest absolute value of $K$ components in $A^\mathrm{T}y$. The $i$th operator: A random selection of $q$ ($K \leq q \leq \mathrm{spark}(A)$) elements in $L$ forms the $c_i$. The operator $\alpha i(0)$ is the set of indicators corresponding to the $K$ components with the largest absolute value in $A_{c_i}^+ y$.

At this point, the initial population is generated. In AOA algorithm, population initialization is carried out randomly. In this paper, the selection of the first operator adopts the simple greedy tracking threshold method [22], When the sparsity of the signal to be restored is small, it can always accurately estimate the support of the signal to be restored. The positions of the remaining operators are the results obtained by using the greedy tracking threshold method for the random components, creating conditions for the search and fast exploitation of the population in space. Further it is necessary to initialize the individual and group optimal positions. The optimal position $S_{i,best}(0)$ of the $i$th operator is set to $\alpha_i(0)$, and the population optimal position is set to $S_{gbest} = \underset{\alpha_i(0)}{\mathrm{argmin}} f(\alpha_i(0)), (i = 1, 2, \ldots, Q)$.

### 3.3. Operator Position Update Mechanism

The position update of the operator in the AOA algorithm is carried out through its own experience and the neighborhood experience. In this paper, the algorithm AOA-CS is based on the population evolution strategy and process, and incorporates a random component to perform location updates [22–26]. Each time the operator starts searching, it first updates *MOA* and *MOP* values, and then generates three random numbers $r1$, $r2$ and $r3$ that obey uniform distribution between 0 and 1. Operator position initialization and group-optimal position initialization have been completed in Section 3.2. During the trajectory of repetition, Division (*D*), Multiplication (*M*), Subtraction (*S*) and Addition (*A*) estimate the feasible positions of the near-optimal solution. Each solution renews its positions from the best-obtained solution. To emphasize exploration and exploitation, the parameter *MOA* is increased linearly from 0.2 to 0.9. Candidate solutions seek to diverge from the near-optimal solution when $r1 > MOA$ and converge towards the near-optimal solution when $r1 < MOA$. The overall position update mechanism is as follow:

When $r1 > MOA$, if $r2 > 0.5$, the $j$ th position of the $i$ th solution is updated by the division operation in Equation(1), if $r2 < 0.5$, the $j$th position of the $i$ th solution is updated by the multiplication operator. When $r1 < MOA$, if $r3 > 0.5$, the $j$ th position of the $i$th solution is updated by the division operation in Equation (2), if $r3 < 0.5$, the $j$th position of the $i$ th solution is updated by the multiplication operator in Equation (2). The current position of the $i$th operator iteration is $S_{i,t}$ $(1 \leq i \leq Q)$ and the historical best position is $S_{i,best}$. After all positions of the $i$ th solution are updated and before the start of the $(t+1)$th iteration, the fitness value of the current number of iterations of the $i$ th operator is calculated by Equation (8). If the operator fitness is less than its individual optimal fitness, namely $f(S_{i,t}) < f(S_{i,best})$, then update the individual optimal position $S_{i,best} = S_{i,t}$, otherwise it remains unchanged. If the individual optimal fitness is less than the population optimal fitness, namely $f(S_{i,best}) < f(S_{gbest})$, then update the population optimal position $S_{gbest} = S_{i,best}$, otherwise it remains unchanged.

The algorithm has been introduced so far, and the flow chart of the algorithm and the flow chart of the mechanical vibration signal reconstruction is shown in Figure 2.

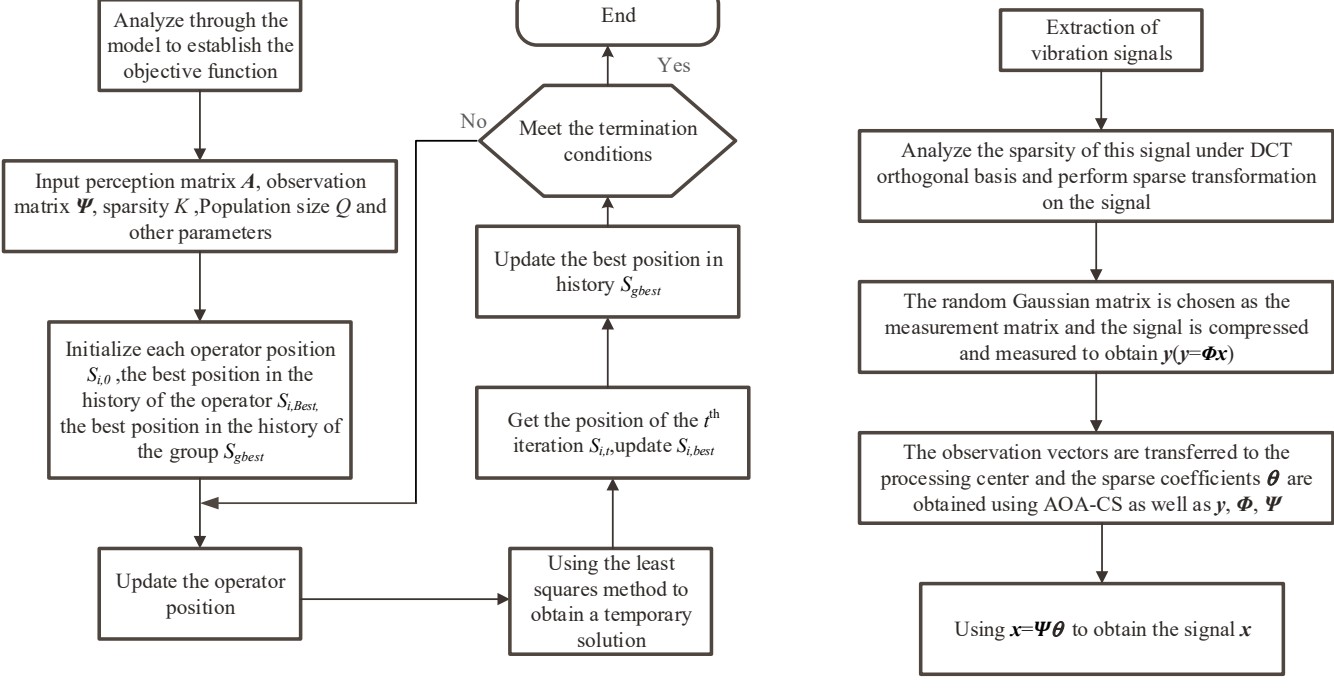

a.Flowchart of the AOA-CS

b.Flowchart of mechanical vibration signal reconstruction

**Figure 2.** Flowchart of optimization algorithm and reconstructed signal.

## 4. Experiment and Analysis

In this section, we experimentally investigate the performance of the proposed algorithm and compare it with the basis pursuit algorithm (BP) [27,28], a typical algorithm in convex optimization, and the orthogonal matching tracking algorithm (OMP) [29,30], a typical algorithm in greedy algorithms. The experimental data in this paper comes from the bearing database of Case Western Reserve University, the object of this experiment is deep groove ball bearing, the sensors are installed in the drive end and fan end respectively for fault data collection, SKF620 is the drive end bearing, SKF6203 is the fan end bearing. The experiment uses acceleration sensors for vibration signal acquisition, including normal data, bearing inner and outer ring fault data, ball fault data, sampling frequency 48 kHz and 12 kHz, the size of the fault diameter is different, respectively 0.018, 0.036, 0.053 CHI,

etc. The state load of the fault is divided into 0, 1, 2, 3 HP (1 HP = 746 W). In this experiment, we select the experimental data of 0.018 cm outer ring diameter of the drive end bearing with a bearing load of 0 and a sampling frequency of 12 kHz. We randomly draw data from this dataset each time to ensure its generalizability.

### 4.1. Sparsity Analysis of Mechanical Vibration Signals

As shown in Figure 3, the distribution of the coefficients of the vibration signal transformed under the DCT orthogonal basis is arranged in descending order from the largest to the smallest in absolute value. As can be seen from the figure: the sparse coefficients of mechanical vibration signals under the DCT orthogonal basis all show an obvious trend of decay, and the slope of the coefficient decay curve decreases sharply and is close to zero after several experiments, so the experimental sparsity K is estimated to be 200. This experiment further verifies that the mechanical vibration signal is compressible under the DCT quadrature basis.

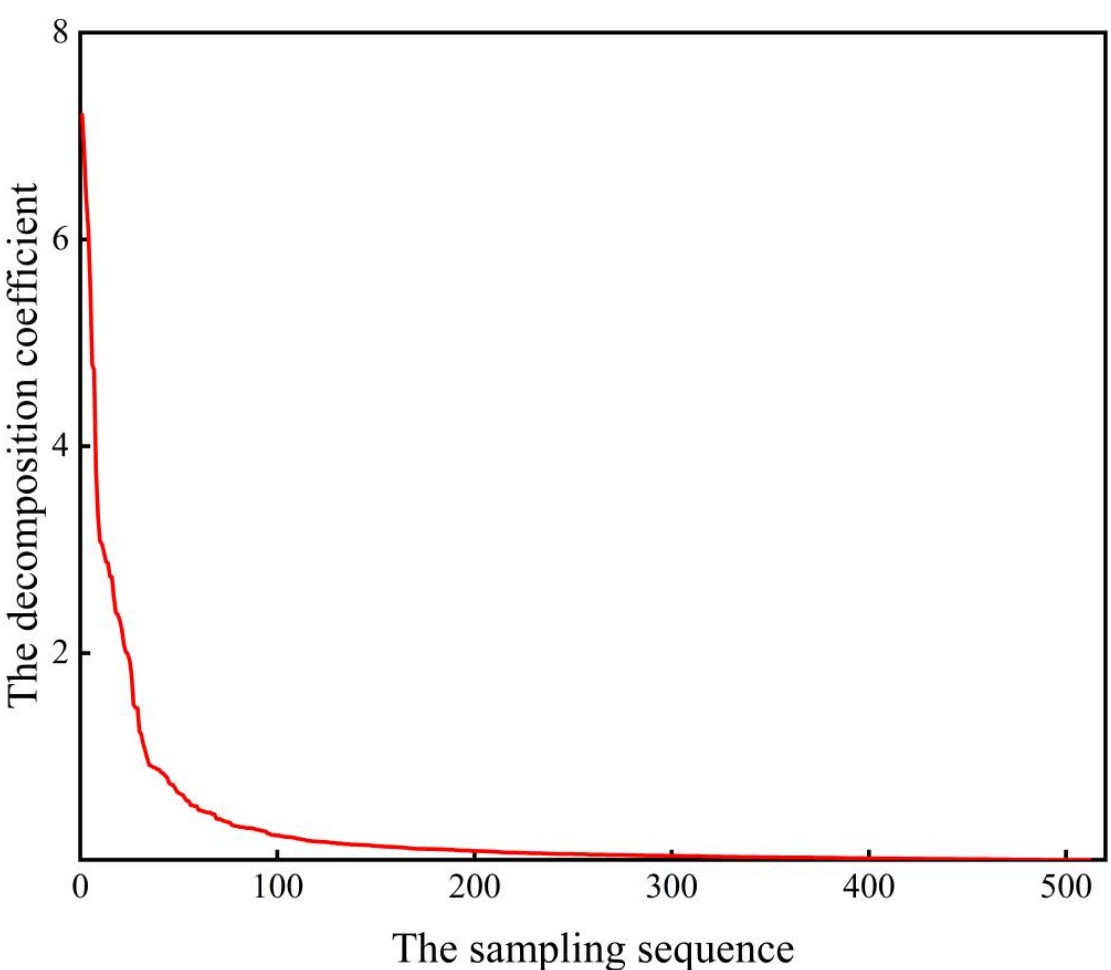

**Figure 3.** The decay distribution of the transformation coefficient of the mechanical vibration signal under the DCT base.

The experiments in this paper use the compression rate to measure the compressibility of mechanical vibration signals, which is defined as follows:

Compression Rate (CR): Indicates the compressibility of the vibration signal. The larger the compression rate, the fewer measurements are required and the more compressible the vibration signal is.

$$R_C = \frac{N - M}{N} \times 100\% \tag{12}$$

where $N$ is the length of the original mechanical vibration signal, $M$ denotes the length of the signal after compression. In order to ensure the high probability reconstruction of the original vibration signal, the compressed measurement number $M$ must satisfy the following inequalitys.

$$M \geq K\lg(\frac{N}{K}) \tag{13}$$

When $N = 512$ and K = 200, we can get $M \geq 82$ by substituting into Equation (13), and $R_C \leq 85\%$ by combining Equations (12) and (13). It can be seen that: mechanical vibration signal compression sampling, the compression rate should not exceed 85% at most, otherwise when the number of measurements is too small, the original vibration signal can not be accurately reconstructed phenomenon.

And when the compression rate is too small, the number of measurements is too many, and there is no meaning for compressive sampling. The analysis shows that the compression rate cannot be too large or too small, so this experiment sets the compression rate to take a range of $50\% \leq R_C \leq 85\%$. The corresponding sampling value range is 82 to 260.

When no noise is present, this paper uses the relative error defined by Equation (14) to measure the recovery performance of the mechanical vibration signal.

$$\sigma = \frac{\left\|\bar{x} - x\right\|_2}{\|x\|_2} \tag{14}$$

where: $\bar{x}$ is the recovered signal, $x$ is the original signal. The smaller the relative error is, the more accurate the reconstruction is. When noise is present, this paper uses the mean square error (MSE) defined by Equation (15) to measure the recovery performance of the mechanical vibration signal.

$$E_{\mathrm{MS}} = \frac{\left\|\bar{x} - x\right\|_2^2}{\|x\|_2^2} \tag{15}$$

where the smaller the mean square error, the better the signal recovery.

All data for the experiments in this paper were run through MATLABR2021a software on a 16 G running memory, dual-core desktop computer. The experimental results are the average of 50 independent experiments.

### 4.2. AOA-CS Algorithm Performance Analysis

Since too high or too low compression rate will have an impact on the original signal compression sampling, storage and transmission, and reconstruction accuracy, this experiment selects the compression rate fixed at 60% and the number of measurements $M = 200$ to verify the reconstruction relative errors and recovery waveform plots of different algorithms for mechanical vibration signals, and the experimental results are shown in Table 1 and Figure 4.

**Table 1.** Comparison of Reconstruction Performance of Different Algorithms with Fixed Compression Rate.

| Algorithm | AOA-CS | OMP | BP |
|-----------|--------|-----|-----|
| Relative error | 0.1038 | 0.1887 | 0.1490 |

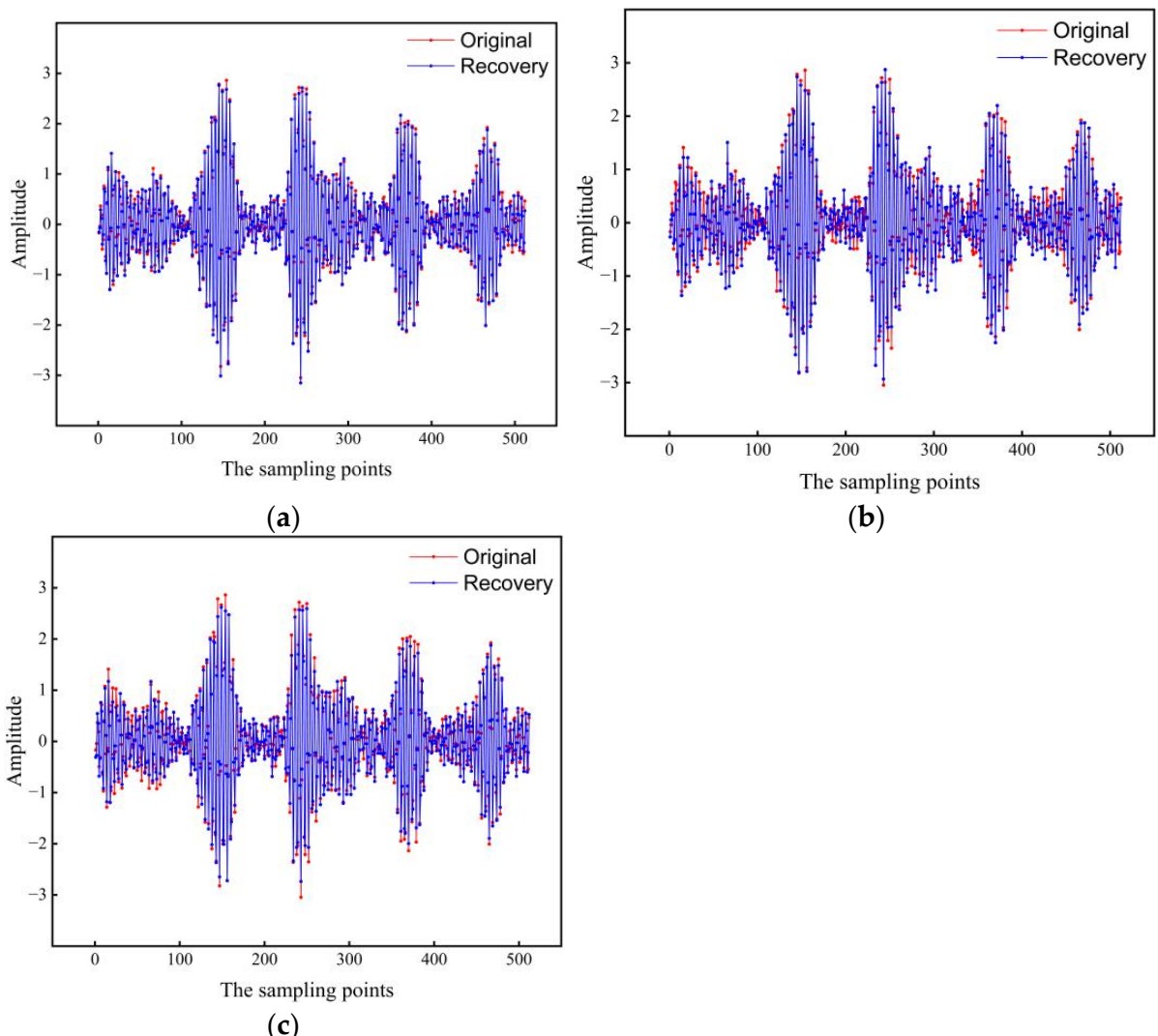

**Figure 4.** AOA-CS Algorithm to recover the mechanical vibration signal compared with the original vibration signal. (**a**) AOA-CS; (**b**) OMP; (**c**) BP.

As can be seen from Table 1, when the compression rate is set at 60%, the reconstruction error of OMP algorithm is the largest, and the reconstruction error of AOA-CS proposed in this paper is the smallest, only 55% of that of OMP algorithm.

It is intuitively seen from Figure 4: when the fixed compression rate is 60%, the mechanical vibration signal recovered by using the proposed algorithm AOA-CS in this paper has the smallest difference with the original vibration signal and is almost perfectly reconstructed, which is consistent with the results of the reconstruction relative error of the vibration signal recovered by different algorithms in Table 1. Therefore, combined with Figure 4 and Table 1, it can be seen that: the proposed algorithm AOA-CS mechanical vibration signal has the best recovery effect, the smallest reconstruction relative error and better adaptability.

### 4.3. Performance Analysis a of AOA-CS Algorithm When the Measured Value Changes

This experiment sets the range of measurement value variation to $85 \leq M \leq 260$ to verify the reconstruction performance of AOA-CS algorithm with the variation of measurement value. The experimental results are shown in Figures 5 and 6.

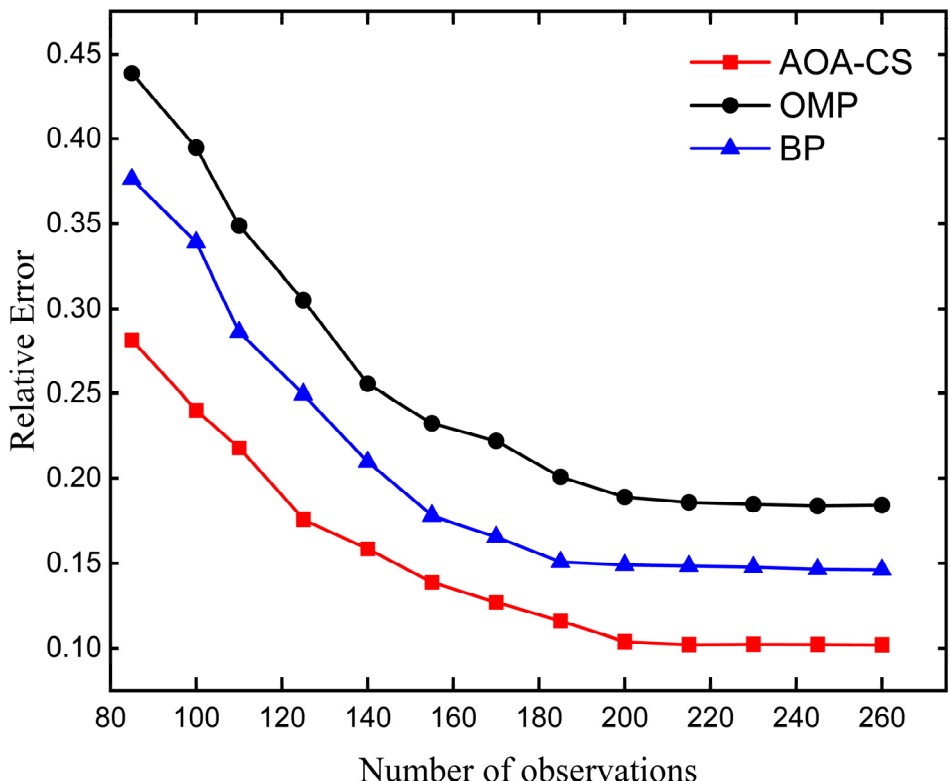

**Figure 5.** Relative error of reconstruction of each algorithm when the measured value changes.

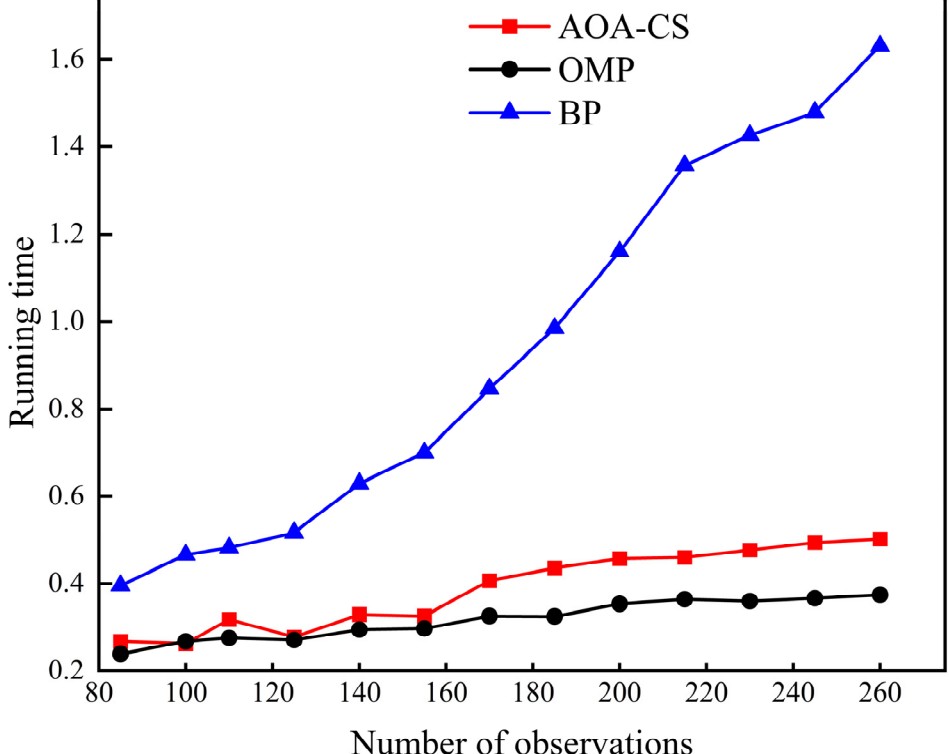

**Figure 6.** Average running time of each algorithm under different observation degrees.

From Figure 5, it can be seen that the relative error of reconstruction of different algorithms in both sets of experiments is decreasing as the measured value increases, i.e., when the compression rate decreases. Among them, the reconstruction relative error

of OMP is the largest, the reconstruction relative error of BP algorithm is the second, the reconstruction relative error of the proposed algorithm AOA-CS algorithm is the smallest, and the reconstruction effect for mechanical vibration signal is the best. And when the measured value is greater than 200, the trend of curve changes of different algorithms are no longer obvious and close to the minimum value, which is consistent with the aforementioned analysis of the range of changes of measured values: when the number of measured values is too large, compression does not have any meaning, and the number of measured values is too small, the original mechanical vibration signal cannot be accurately reconstructed.

From Figure 6, it can be seen that the proposed algorithm is more capable of recovering the vibration signal and has higher computational efficiency than the BP algorithm. It is worth mentioning that the algorithm in this paper has a natural parallelism capability, which makes it possible to run on a distributed processor and obtain a greater degree of computational efficiency.

### 4.4. Performance Analysis a of AOA-CS Algorithm wh in the Presence of Noise

When the signal-to-noise ratio (SNR) is high, the acquired signal contains more useful information, while when the signal-to-noise ratio (SNR) is low, it contains less useful information. Therefore, when there is noise, the SNR will have some influence on the recovery performance of the signal. This experiment assumes that the noise is Gaussian white noise, and sets the SNR to be 5 to 35 dB to verify the recovery performance of the AOA-CS algorithm for mechanical vibration signals when noise exists. The experimental results are shown in Figures 7 and 8.

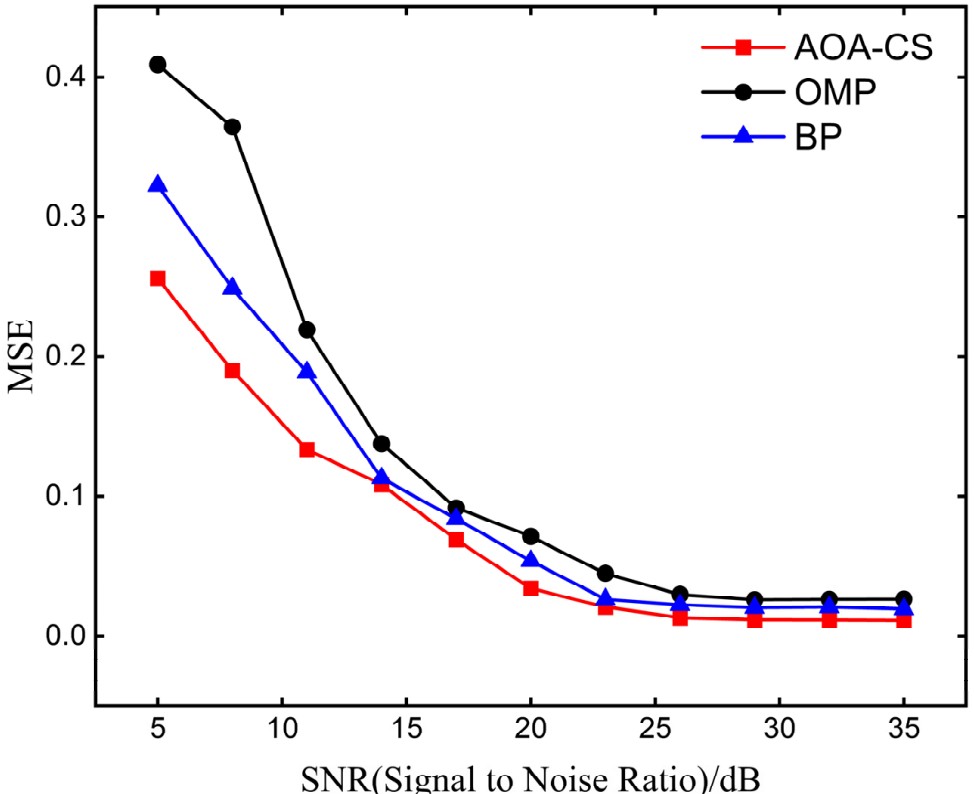

**Figure 7.** Reconstructed mean square error of AOA-CS algorithm in the presence of noise.

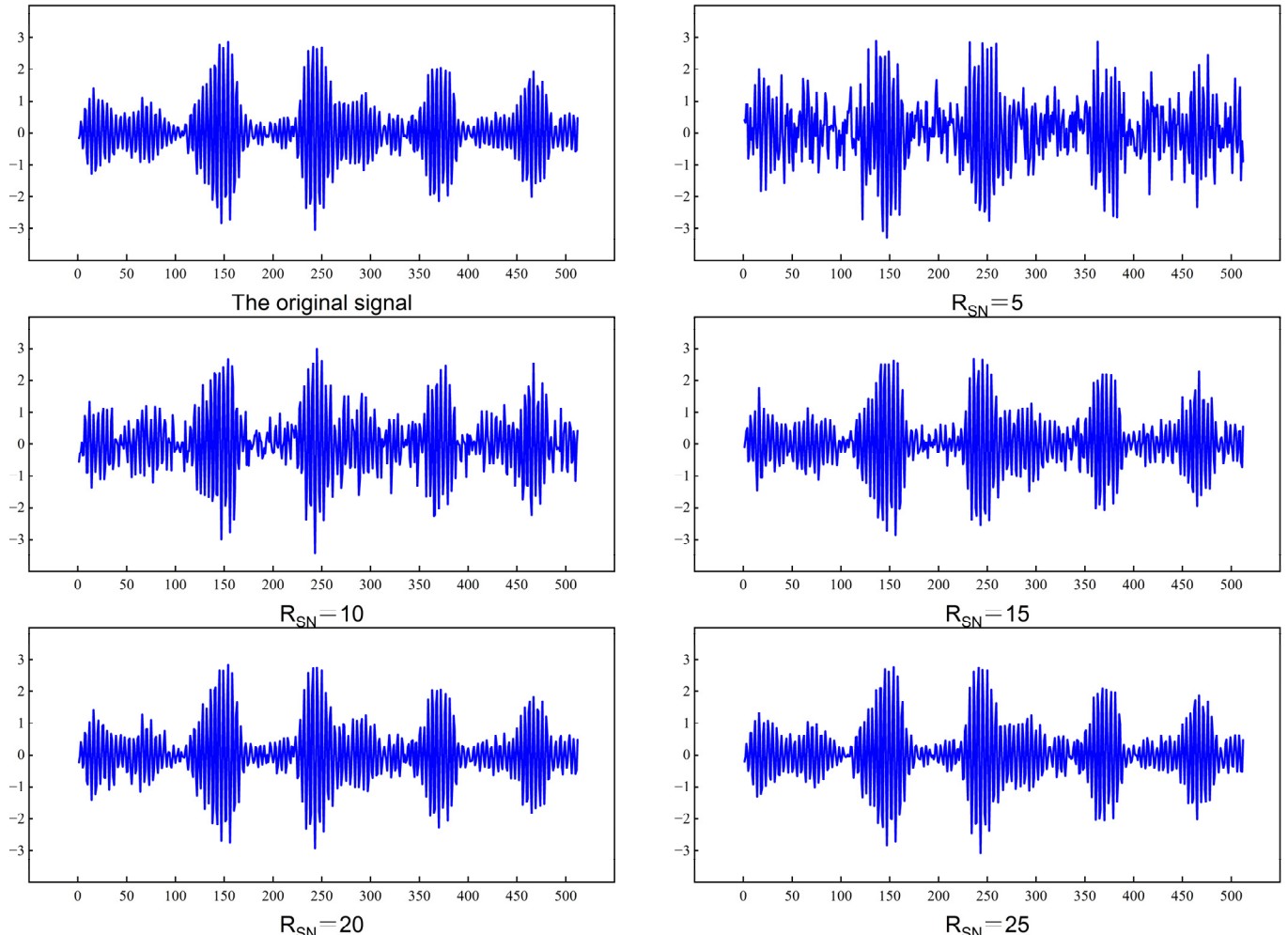

**Figure 8.** Aveforms of mechanical vibration signals at different signal-to-noise ratios.

Figure 7 shows that when noise is present, the reconstruction mean square error of different algorithms in both sets of experiments decreases as the SNR increases. Among them, OMP algorithm has the largest reconstruction mean square error, BP algorithm has the second largest mean square error, AOA-CS algorithm in this paper has the smallest reconstruction mean square error, i.e., the most accurate recovery. And when $R_{SN} > 20$ dB, the reconstruction mean square error variation is no longer significant and almost reaches the minimum value. As shown in Figure 7 is the mechanical vibration signal wave signal diagram in the presence of noise.

As can be seen from Figure 8: With the increase of SNR, the waveform diagram of mechanical vibration signal is gradually close to that of the original vibration signal. When the signal-to-noise ratio is 5 dB, the difference between the waveform and the original vibration signal waveform is the greatest, which is due to the fact that the signal contains less useful information when the signal-to-noise ratio is low, when $R_{SN} = 10$ dB and $R_{SN} = 15$ dB, the waveform gradually becomes similar to the original vibration signal waveform, when $R_{SN} = 20$ dB, the waveform is almost the same as the original vibration signal waveform. When the signal-to-noise ratio is $R_{SN} = 25$ dB, the original vibration signal is also almost reproduced. This is due to the fact that the signal contains more useful information when the signal-to-noise ratio is higher, i.e., it contains more information of the original signal, which is more conducive to the reconstruction recovery of the signal containing noise at this time, which is consistent with the results of the graph of the mechanical vibration signal with noise shown in Figure 7.

## 5. Conclusions

This paper proposes a sparse recovery algorithm based on arithmetic optimization algorithm, combined with the idea of greedy tracking method. Through compressed sensing model and formula derivation, this paper introduces arithmetic optimization algorithm to solve the sparse coefficient of signal in its sparse domain, and designs the initial position of the whole group in combination with the cutting technique of greedy tracking method, which solves the problem that traditional reconstruction algorithms cannot take into account the accuracy and efficiency of reconstruction. At the same time, it improves the anti-interference of the whole reconstruction process to a certain extent. On the one hand, the algorithm inherits the global search feature of arithmetic optimization and has stronger recovery ability. On the other hand, the algorithm takes advantage of the fast and effective greedy tracking method and can terminate quickly under the general sparsity and observation conditions. The results of various numerical experiments show that:

(1) As the number of measurements increases, i.e., the compression rate decreases, the recovery reconstruction error of each algorithm becomes smaller and smaller, but the proposed method in this paper always has optimal performance.

(2) The proposed method also has the minimum mean square error of reconstruction in the presence of noise.

The algorithm proposed in this paper has some room for improvement in terms of operator position updating. Further research will be conducted to achieve better results. This work makes sense for the monitoring systems for mechanical equipment, since the proposed scheme can provide more information for fault diagnosis.

If there is a need for the code can contact the author, our code of programs and example is availiable upon request.

**Author Contributions:** Conceptualization, Q.Z. and J.X., methodology, Q.Z., software, Q.Z. and D.H., validation, Q.Z., J.X., D.H. and C.T., formal analysis, Q.Z., J.X. and D.H., investigation, Q.Z., D.H. and C.T., writing—original draft preparation, Q.Z., writing—review and editing, Q.Z., J.X., D.H. and C.T., visualization, Q.Z., D.H. and C.T., supervision, J.X. and C.T., funding acquisition, Q.Z, J.X., D.H. and C.T. All authors have read and agreed to the published version of the manuscript.

**Funding:** This work was supported by the National Natural Science Foundation of China (No.51977179).

**Data Availability Statement:** If there is a need for the code can contact the author, our code of programs and example is availiable upon request.

**Conflicts of Interest:** The authors declare no conflict of interest.

## Abbreviations

The following abbreviations are used in this manuscript.

| | |
|---|---|
| CS | Compressive sensing |
| AOA | Arithmetic Optimization Algorithm |
| M | Multiplication |
| D | Division |
| S | Subtraction |
| A | Addition |
| DCT | Discrete cosine orthogonal basis |
| BP | Basis pursuit |
| OMP | Orthogonal matching pursuit |
| CR | Compression Rate |
| SNR | Signal-to-noise ratio |
| MSE | Mean square error |

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
