# Peer review of "A Sparse Recovery Algorithm Based on Arithmetic Optimization"

_electronics, doi:10.3390/electronics12010162_

Round 1

Reviewer 1 Report

Please open the attachment file.

Reviewer 2 Report

The paper presents an interesting approach to sampling and reconstructing mechanical vibration signals using compressed sensing theory. The authors argue that traditional Nyquist sampling is not effective for dealing with the complex, non-stationary signals produced by large equipment, and propose the use of compressed sensing as an alternative. The paper provides a detailed description of the recovery model based on the l0 norm, and discusses the advantages of this approach over traditional methods. The experiments are strong to cover the claim.

Reviewer 3 Report

The manuscript A 'Sparce Recovery Algorithm Based on Arithmetic Optimization' has been submitted to then journal 'MDPI Electronics.' The authors propose a new sparse recovery algorithm. The authors show that compared with some other methods, the proposed algorithm produces more effective recovery and runs faster under some general conditions. Also, the method can recover the signal better in the presence of noise.

I have the following comments:

The results are important from the practical point of view. Efficient and accurate algorithms are needed, for example, in signal processing applications.

1) The paper needs extensive editing of the text and layout of the paper.

2) The use of space (" "), ";", ",". You should use "," before "where" and begin the next line without extra intention. Correct the format of titles.

3) Correct references 1-5. Ieee->IEEE. Reference 14 don't use capital letters.

4) You could add a list of abbreviations.

5) M_Iter denotes... repetition lines 105, 111 and 116.

6) corresponding to the corresponding to the... line 181

7) D, M, SW, and A line 205?

8) Consistent use of "Fig." "Figure"

Round 2

Reviewer 3 Report

The authors have addressed all my questions and concerns.